# Deep Anomaly Detection Using Geometric Transformations

**Izhak Golan**
Department of Computer Science
Technion – Israel Institute of Technology
Haifa, Israel
izikgo@cs.technion.ac.il

**Ran El-Yaniv**
Department of Computer Science
Technion – Israel Institute of Technology
Haifa, Israel
rani@cs.technion.ac.il

## Abstract

We consider the problem of anomaly detection in images, and present a new detection technique. Given a sample of images, all known to belong to a "normal" class (e.g., dogs), we show how to train a deep neural model that can detect out-of-distribution images (i.e., non-dog objects). The main idea behind our scheme is to train a multi-class model to discriminate between dozens of geometric transformations applied on all the given images. The auxiliary expertise learned by the model generates feature detectors that effectively identify, at test time, anomalous images based on the softmax activation statistics of the model when applied on transformed images. We present extensive experiments using the proposed detector, which indicate that our technique consistently improves all known algorithms by a wide margin.

## 1   Introduction

Future machine learning applications such as self-driving cars or domestic robots will, inevitably, encounter various kinds of risks including statistical uncertainties. To be usable, these applications should be as robust as possible to such risks. One such risk is exposure to statistical errors or inconsistencies due to distributional divergences or noisy observations. The well-known problem of *anomaly/novelty detection* highlights some of these risks, and its resolution is of the utmost importance to mission critical machine learning applications. While anomaly detection has long been considered in the literature, conclusive understanding of this problem in the context of deep neural models is sorely lacking. For example, in machine vision applications, presently available novelty detection methods can suffer from poor performance in some problems, as demonstrated by our experiments.

In the basic anomaly detection problem, we have a sample from a "normal" class of instances, emerging from some distribution, and the goal is to construct a classifier capable of detecting out-of-distribution "abnormal" instances [5].[1] There are quite a few variants of this basic anomaly detection problem. For example, in the *positive and unlabeled* version, we are given a sample from the "normal" class, as well as an unlabeled sample that is contaminated with abnormal instances. This contaminated-sample variant turns out to be easier than the pure version of the problem (in the sense that better performance can be achieved) [2]. In the present paper, we focus on the basic (and harder) version of anomaly detection, and consider only machine vision applications for which deep models (e.g., convolutional neural networks) are essential.

There are a few works that tackle the basic, pure-sample-anomaly detection problem in the context of images. The most successful results among these are reported for methods that rely on one of

the following two general schemes. The first scheme consists of methods that analyze errors in reconstruction, which is based either on autoencoders or generative adversarial models (GANs) trained over the normal class. In the former case, reconstruction deficiency of a test point indicates abnormality. In the latter, the reconstruction error of a test instance is estimated using optimization to find the approximate inverse of the generator. The second class of methods utilizes an autoencoder trained over the normal class to generate a low-dimensional embedding. To identify anomalies, one uses classical methods over this embedding, such as low-density rejection [8, 9] or single-class SVM [29, 30]. A more advanced variant of this approach combines these two steps (encoding and then detection) using an appropriate cost function, which is used to train a single neural model that performs both procedures [27].

In this paper we consider a completely different approach that bypasses reconstruction (as in autoencoders or GANs) altogether. The proposed method is based on the observation that learning to discriminate between many types of geometric transformations applied to normal images, encourages learning of features that are useful for detecting novelties. Thus, we train a multi-class neural classifier over a *self-labeled* dataset, which is created from the normal instances and their transformed versions, obtained by applying numerous geometric transformations. At test time, this discriminative model is applied on transformed instances of the test example, and the distribution of softmax response values of the "normal" train images is used for effective detection of novelties. The intuition behind our method is that by training the classifier to distinguish between transformed images, it must learn salient geometrical features, some of which are likely to be unique to the single class.

We present extensive experiments of the proposed method and compare it to several state-of-the-art methods for pure anomaly detection. We evaluate performance using a one-vs-all scheme over several image datasets such as CIFAR-100, which (to the best of our knowledge) have never been considered before in this setting. Our results overwhelmingly indicate that the proposed method achieves dramatic improvements over the best available methods. For example, on the CIFAR-10 dataset (10 different experiments), we improved the top performing baseline AUROC by 32% on average. In the CatsVsDogs dataset, we improve the top performing baseline AUROC by 67%.

## 2 Related Work

The literature related to anomaly detection is extensive and beyond the scope of this paper (see, e.g., [5, 42] for wider scope surveys). Our focus is on anomaly detection in the context of images and deep learning. In this scope, most published works rely, implicitly or explicitly, on some form of (unsupervised) reconstruction learning. These methods can be roughly categorized into two approaches.

**Reconstruction-based anomaly score.**   These methods assume that anomalies possess different visual attributes than their non-anomalous counterparts, so it will be difficult to compress and reconstruct them based on a reconstruction scheme optimized for single-class data. Motivated by this assumption, the anomaly score for a new sample is given by the quality of the reconstructed image, which is usually measured by the $\ell_2$ distance between the original and reconstructed image. Classic methods belonging to this category include Principal Component Analysis (PCA) [18], and Robust-PCA [4]. In the context of deep learning, various forms of deep autoencoders are the main tool used for reconstruction-based anomaly scoring. Xia et al. [37] use a convolutional autoencoder with a regularizing term that encourages outlier samples to have a large reconstruction error. Variational autoencoder is used by An and Cho [1], where they estimate the reconstruction probability through Monte-Carlo sampling, from which they extract an anomaly score. Another related method, which scores an unseen sample based on the ability of the model to generate a similar one, uses Generative Adversarial Networks (GANS) [16]. Schlegl et al. [28] use this approach on optical coherence tomography images of the retina. Deecke et al. [7] employ a variation of this model called ADGAN, reporting slightly superior results on CIFAR-10 [21] and MNIST [22].

**Reconstruction-based representation learning.**   Many conventional anomaly detection methods use a low-density rejection principle [8]. Given data, the density at each point is estimated, and new samples are deemed anomalous when they lie in a low-density region. Examples of such methods are kernel density estimation (KDE) [25], and Robust-KDE [19]. This approach is known to be problematic when handling high-dimensional data due to the *curse of dimensionality*. To mitigate

this problem, practitioners often use a two-step approach of learning a compact representation of the data, and then applying density estimation methods on the lower-dimensional representation [4]. More advanced techniques combine these two steps and aim to learn a representation that facilitates the density estimation task. Zhai et al. [41] utilize an energy-based model in the form of a regularized autoencoder in order to map each sample to an energy score, which is the estimated negative log-probability of the sample under the data distribution. Zong et al. [43] uses the representation layer of an autoencoder in order to estimate parameters of a Gaussian mixture model.

There are few approaches that tackled the anomaly detection problem without resorting to some form of reconstruction. A recent example was published by Ruff et al. [27], who have developed a deep one-class SVM model. The model consists of a deep neural network whose weights are optimized using a loss function resembling the SVDD [30] objective.

## 3   Problem Statement

In this paper, we consider the problem of anomaly detection in images. Let $\mathcal{X}$ be the space of all "natural" images, and let $X \subseteq \mathcal{X}$ be the set of images defined as *normal*. Given a sample $S \subseteq X$, and a type-II error constraint (rate of normal samples that were classified as anomalies), we would like to learn the best possible (in terms of type-I error) classifier $h_S(x) : \mathcal{X} \to \{0, 1\}$, where $h_S(x) = 1 \Leftrightarrow x \in X$, which satisfies the constraint. Images that are not in $X$ are referred to as *anomalies* or *novelties*.

To control the trade-off between type-I and type-II errors when classifying, a common practice is to learn a scoring (ranking) function $n_S(x) : \mathcal{X} \to \mathbb{R}$, such that higher scores indicate that samples are more likely to be in $X$. Once such a scoring function has been learned, a classifier can be constructed from it by specifying an anomaly threshold ($\lambda$):

$$h_S^\lambda(x) = \begin{cases} 1 & n_S(x) \geq \lambda \\ 0 & n_S(x) < \lambda. \end{cases}$$

As many related works [28, 31, 17], in this paper we also focus only on learning the scoring function $n_S(x)$, and completely ignore the constrained binary decision problem. A useful (and common practice) performance metric to measure the quality of the trade-off of a given scoring function is the area under the Receiver Operating Characteristic (ROC) curve, which we denote here as AUROC. When prior knowledge on the proportion of anomalies is available, the area under the precision-recall curve (AUPR) metric might be preferred [6]. We also report on performance in term of this metric in the supplementary material.

## 4   Discriminative Learning of an Anomaly Scoring Function Using Geometric Transformations

As noted above, we aim to learn a scoring function $n_S$ (as described in Section 3) in a discriminative fashion. To this end, we create a self-labeled dataset of images from our initial training set $S$, by using a class of geometric transformations $\mathcal{T}$. The created dataset, denoted $S_\mathcal{T}$, is generated by applying each geometric transformation in $\mathcal{T}$ on all images in $S$, where we label each transformed image with the index of the transformation that was applied on it. This process creates a self-labeled multi-class dataset (with $|\mathcal{T}|$ classes) whose cardinality is $|\mathcal{T}||S|$. After the creation of $S_\mathcal{T}$, we train a multi-class image classifier whose objective is to predict, for each image, the index of its generating transformation in $\mathcal{T}$. At inference time, given an unseen image $x$, we decide whether it belongs to the normal class by first applying each transformation on it, and then applying the classifier on each of the $|\mathcal{T}|$ transformed images. Each such application results in a softmax response vector of size $|\mathcal{T}|$. The final normality score is defined using the combined log-likelihood of these vectors under an estimated distribution of "normal" softmax vectors (see details below).

### 4.1   Creating and Learning the Self-Labeled Dataset

Let $\mathcal{T} = \{T_0, T_1, \dots, T_{k-1}\}$ be a set of geometric transformations, where for each $1 \leq i \leq k-1, T_i : \mathcal{X} \to \mathcal{X}$, and $T_0(x) = x$ is the identity transformation. The set $\mathcal{T}$ is a hyperparameter of our method, on which we elaborate in Section 6. The self-labeled set $S_\mathcal{T}$ is defined as

$$S_\mathcal{T} \triangleq \{(T_j(x), j) : x \in S, T_j \in \mathcal{T}\}.$$

Thus, for any $x \in S$, $j$ is the label of $T_j(x)$. We use this set to straightforwardly learn a deep $k$-class classification model, $f_\theta$, which we train over the self-labeled dataset $S_\mathcal{T}$ using the standard cross-entropy loss function. To this end, any useful classification architecture and optimization method can be employed for this task.

## 4.2 Dirichlet Normality Score

We now define our normality score function $n_S(x)$. Fix a set of geometric transformations $\mathcal{T} = \{T_0, T_1, \dots, T_{k-1}\}$, and assume that a $k$-class classification model $f_\theta$ has been trained on the self-labeled set $S_\mathcal{T}$ (as described above). For any image $x$, let $\mathbf{y}(x) \triangleq \text{softmax}\,(f_\theta\,(x))$, i.e., the vector of softmax responses of the classifier $f_\theta$ applied on $x$. To construct our normality score we define:

$$n_S(x) \triangleq \sum_{i=0}^{k-1} \log p(\mathbf{y}(T_i(x))|T_i),$$

which is the combined log-likelihood of a transformed image conditioned on each of the applied transformations in $\mathcal{T}$, under a naïve (typically incorrect) assumption that all of these conditional distributions are independent. We approximate each conditional distribution to be $\mathbf{y}(T_i(x))|T_i \sim \text{Dir}(\boldsymbol{\alpha}_i)$, where $\boldsymbol{\alpha}_i \in \mathbb{R}^k_+$, $x \sim p_X(x)$, $i \sim \text{Uni}(0, k-1)$, and $p_X(x)$ is the real data probability distribution of "normal" samples. Our choice of the Dirichlet distribution is motivated by two reasons. First, it is a common choice for distribution approximation when samples (i.e., $\mathbf{y}$) reside in the unit $k-1$ simplex. Second, there are efficient methods for numerically estimating the maximum likelihood parameters [24, 34]. We denote the estimation by $\tilde{\boldsymbol{\alpha}}_i$. Using the estimated Dirichlet parameters, the normality score of an image $x$ is:

$$n_S(x) = \sum_{i=0}^{k-1} \left[ \log \Gamma(\sum_{j=0}^{k-1} [\tilde{\boldsymbol{\alpha}}_i]_j) - \sum_{j=0}^{k-1} \log \Gamma([\tilde{\boldsymbol{\alpha}}_i]_j) + \sum_{j=0}^{k-1} ([\tilde{\boldsymbol{\alpha}}_i]_j - 1) \log \mathbf{y}(T_i(x))_j \right].$$

Since all $\tilde{\boldsymbol{\alpha}}_i$ are constant w.r.t $x$, we can ignore the first two terms in the parenthesis and redefine a simplified normality score, which is equivalent in its normality ordering:

$$n_S(x) = \sum_{i=0}^{k-1} \sum_{j=0}^{k-1} ([\tilde{\boldsymbol{\alpha}}_i]_j - 1) \log \mathbf{y}(T_i(x))_j = \sum_{i=0}^{k-1} (\tilde{\boldsymbol{\alpha}}_i - 1) \cdot \log \mathbf{y}(T_i(x)).$$

As demonstrated in our experiments, this score tightly captures normality in the sense that for two images $x_1$ and $x_2$, $n_S(x_1) > n_S(x_2)$ tend to imply that $x_1$ is "more normal" than $x_2$. For each $i \in \{0, \dots, k-1\}$, we estimate $\tilde{\boldsymbol{\alpha}}_i$ using the fixed point iteration method described in [24], combined with the initialization step proposed by Wicker et al. [34]. Each vector $\tilde{\boldsymbol{\alpha}}_i$ is estimated based on the set $S_i = \{\mathbf{y}(T_i(x))|x \in S\}$. We note that the use of an independent image set for estimating $\tilde{\boldsymbol{\alpha}}_i$ may improve performance. A full and detailed algorithm is available in the supplementary material.

A simplified version of the proposed normality score was used during preliminary stages of this research: $\hat{n}_S(x) \triangleq \frac{1}{k} \sum_{j=0}^{k-1} [\mathbf{y}\,(T_j(x))]_j$. This simple score function eliminates the need for the Dirichlet parameter estimation, is easy to implement, and still achieves excellent results that are only slightly worse than the above Dirichlet score.

## 5 Experimental Results

In this section, we describe our experimental setup and evaluation method, the baseline algorithms we use for comparison purposes, the datasets, and the implementation details of our technique (architecture used and geometric transformations applied). We then present extensive experiments on the described publicly available datasets, demonstrating the effectiveness of our scoring function. Finally, we show that our method is also effective at identifying out-of-distribution samples in labeled multi-class datasets.

### 5.1 Baseline Methods

We compare our method to state-of-the-art deep learning approaches as well as a few classic methods.

**One-Class SVM.** The one-class support vector machine (OC-SVM) is a classic and popular kernel-based method for novelty detection [29, 30]. It is typically employed with an RBF kernel, and learns a collection of closed sets in the input space, containing most of the training samples. Samples residing outside of these enclosures are deemed anomalous. Following [41, 7], we use this model on raw input (i.e., a flattened array of the pixels comprising an image), as well as on a low-dimensional representation obtained by taking the bottleneck layer of a trained convolutional autoencoder. We name these models **RAW-OC-SVM** and **CAE-OC-SVM**, respectively. It is very important to note that in both these variants of OC-SVM, we provide the OC-SVM with an **unfair significant advantage** by optimizing its hyperparameters in *hindsight*; i.e., the OC-SVM hyperparameters ($\nu$ and $\gamma$) were optimized to maximize AUROC and taken to be the best performing values among those in the parameter grid: $\nu \in \{0.1, 0.2, \ldots, 0.9\}$, $\gamma \in \{2^{-7}, 2^{-6}, \ldots, 2^2\}$. Note that the hyperparameter optimization procedure has been provided with a two-class classification problem. There are, in fact, methods for optimizing these parameters without hindsight knowledge [33, 3]. These methods are likely to degrade the performance of the OC-SVM models. The convolutional autoencoder is chosen to have a similar architecture to that of DCGAN [26], where the encoder is adapted from the discriminator, and the decoder is adapted from the generator.

In addition, we compare our method to a recently published, end-to-end variant of OC-SVM called **One-Class Deep SVDD** [27]. This model, which we name **E2E-OC-SVM**, uses an objective similar to that of the classic SVDD [30] to optimize the weights of a deep architecture. However, there are constraints on the used architecture, such as lack of bias terms and unbounded activation functions. The experimental setup used by the authors is identical to ours, allowing us to report their published results as they are, on CIFAR-10.

**Deep structured energy-based models.** A deep structured energy-based model (**DSEBM**) is a state-of-the-art deep neural technique, whose output is the energy function (negative log probability) associated with an input sample [41]. Such models can be trained efficiently using score matching in a similar way to a denoising autoencoder [32]. Samples associated with high energy are considered anomalous. While the authors of [41] used a very shallow architecture in their model (which is ineffective in our problems), we selected a deeper one when using their method. The chosen architecture is the same as that of the encoder part in the convolutional autoencoder used by CAE-OC-SVM, with ReLU activations in the encoding layer.

**Deep Autoencoding Gaussian Mixture Model.** A deep autoencoding Gaussian mixture model (**DAGMM**) is another state-of-the-art deep autoencoder-based model, which generates a low-dimensional representation of the training data, and leverages a Gaussian mixture model to perform density estimation on the compact representation [43]. A DAGMM jointly and simultaneously optimizes the parameters of the autoencoder and the mixture model in an end-to-end fashion, thus leveraging a separate estimation network to facilitate the parameter learning of the mixture model. The architecture of the autoencoder we used is similar to that of the convolutional autoencoder from the CAE-OC-SVM experiment, but with linear activation in the representation layer. The estimation network is inspired by the one in the original DAGMM paper.

**Anomaly Detection with a Generative Adversarial Network.** This network, given the acronym ADGAN, is a GAN based model, which learns a one-way mapping from a low-dimensional multi-variate Gaussian distribution to the distribution of the training set [7]. After training the GAN on the "normal" dataset, the discriminator is discarded. Given a sample, the training of ADGAN uses gradient descent to estimate the inverse mapping from the image to the low-dimensional seed. The seed is then used to generate a sample, and the anomaly score is the $\ell_2$ distance between that image and the original one. In our experiments, for the generative model of the ADGAN we incorporated the same architecture used by the authors of the original paper, namely, the original DCGAN architecture [26]. As described, ADGAN requires only a trained generator.

## 5.2 Datasets

We consider four image datasets in our experiments: CIFAR-10, CIFAR-100 [21], CatsVsDogs [11], and fashion-MNIST [38], which are described below. We note that in all our experiments, pixel values of all images were scaled to reside in $[-1, 1]$. No other pre-processing was applied.

- **CIFAR-10:** consists of 60,000 32x32 color images in 10 classes, with 6,000 images per class. There are 50,000 training images and 10,000 test images, divided equally across the classes.

- **CIFAR-100:** similar to CIFAR-10, but with 100 classes containing 600 images each. This set has a fixed train/test partition with 500 training images and 100 test images per class. The 100 classes in the CIFAR-100 are grouped into 20 superclasses, which we use in our experiments.

- **Fashion-MNIST:** a relatively new dataset comprising 28x28 grayscale images of 70,000 fashion products from 10 categories, with 7,000 images per category. The training set has 60,000 images and the test set has 10,000 images. In order to be compatible with the CIFAR-10 and CIFAR-100 classification architectures, we zero-pad the images so that they are of size 32x32.

- **CatsVsDogs:** extracted from the ASIRRA dataset, it contains 25,000 images of cats and dogs, 12,500 in each class. We split this dataset into a training set containing 10,000 images, and a test set of 2,500 images in each class. We also rescale each image to size 64x64. The average dimension size of the original images is roughly 360x400.

## 5.3 Experimental Protocol

We employ a one-vs-all evaluation scheme in each experiment. Consider a dataset with $C$ classes, from which we create $C$ different experiments. For each $1 \leq c \leq C$, we designate class $c$ to be the single class of normal images. We take $S$ to be the set of images in the training set belonging to class $c$. The set $S$ is considered to be the set of "normal" samples based on which the model must learn a normality score function. We emphasize that $S$ contains *only* normal samples, and no additional samples are provided to the model during training. The normality score function is then applied on all images in the test set, containing both anomalies (not belonging to class $c$) and normal samples (belonging to class $c$), in order to evaluate the model's performance. As stated in Section 3, we completely ignore the problem of choosing the appropriate anomaly threshold ($\lambda$) on the normality score, and quantify performance using the area under the ROC curve metric, which is commonly utilized as a performance measure for anomaly detection models. We are able to compute the ROC curve since we have full knowledge of the ground truth labels of the test set.[2]

**Hyperparameters and Optimization Methods** For the self-labeled classification task, we use 72 geometric transformations. These transformations are specified in the supplementary material (see also Section 6 discussing the intuition behind the choice of these transformations). Our model is implemented using the state-of-the-art Wide Residual Network (WRN) model [40]. The parameters for the depth and width of the model for all 32x32 datasets were chosen to be 10 and 4, respectively, and for the CatsVsDogs dataset (64x64), 16 and 8, respectively. These hyperparameters were selected prior to conducting any experiment, and were fixed for all runs.[3] We used the Adam [20] optimizer with default hyperparameters. Batch size for all methods was set to 128. The number of epochs was set to 200 on all benchmark models, except for training the GAN in ADGAN for which it was set to 100 and produced superior results. We trained the WRN for $\lceil 200/|\mathcal{T}| \rceil$ epochs on the self-labeled set $S_{\mathcal{T}}$, to obtain approximately the same number of parameter updates as would have been performed had we trained on $S$ for 200 epochs.

## 5.4 Results

In Table 1 we present our results. The table is composed of four blocks, with each block containing several anomaly detection problems derived from the same dataset (for lack of space we omit class names from the tables, and those can be found in the supplementary material). For example, the first row contains the results for an anomaly detection problem where the normal class is class 0 in CIFAR-10 (*airplane*), and the anomalous instances are images from all other classes in CIFAR-10 (classes 1-9). In this row (as in any other row), we see the average AUROC results over five runs and the corresponding standard error of the mean for all baseline methods. The results of our algorithm are shown in the rightmost column. OC-SVM variants and ADGAN were run once due to their

time complexity. The best performing method in each row appears in bold. For example, in the CatsVsDogs experiments where *dog* (class 1) is the "normal" class, the best baseline (DSEBM) achieves 0.561 AUROC. Note that the trivial average AUROC is always 0.5, regardless of the proportion of normal vs. anomalous instances. Our method achieves an average AUROC of 0.888.

Several interesting observations can be made by inspecting the numbers in Table 1. Our relative advantage is most prominent when focusing on the larger images. All baseline methods, including OC-SVM variants, which enjoy hindsight information, only achieve performance that is slightly better than random guessing in the CatsVsDogs dataset. On the smaller-sized images, the baselines can perform much better. In most cases, however, our algorithm significantly outperformed the other methods. Interestingly, in many cases where the baseline methods struggled with separating normal samples from anomalies, our method excelled. See, for instance, the cases of *automobile* (class 1) and *horse* (class 7; see the CIFAR-10 section in the table). Inspecting the results on CIFAR-100 (where 20 super-classes defined the partition), we observe that our method was challenged by the diversity inside the normal class. In this case, there are a few normal classes on which our method did not perform well; see e.g., *non-insect invertebrates* (class 13), *insects* (class 7), and *household electrical devices* (class 5). In Section 6 we speculate why this might happen. We used the super-class partitioning of CIFAR-100 (instead of the 100 base classes) because labeled data for single base classes is scarce. On the fashion-MNIST dataset, all methods, excluding DAGMM, performed very well, with a slight advantage to our method. The fashion-MNIST dataset was designed as a drop-in replacement for the original MNIST dataset, which is *slightly* more challenging. Classic models, such as SVM with an RBF kernel, can perform well on this task, achieving almost 90% accuracy [38].

## 5.5 Identifying Out-of-distribution Samples in Labeled Multi-class Datasets

Although it is not the main focus of this work, we have also tackled the problem of identifying out-of-distribution samples in labeled multi-class datasets (i.e., identify images that belong to a different distribution than that of the labeled dataset). To this end, we created a two-headed classification model based on the WRN architecture. The model has two separate softmax output layers. One for categories (e.g., cat, truck, airplane, etc.) and another for classifying transformations (our method). We use the categories softmax layer only during training. At test time, we only utilize the transformations softmax layer output as described in section 4.2, but use the simplified normality score. When training on the CIFAR-10 dataset, and taking the tiny-imagenet (resized) dataset to be anomalies as done by Liang et al. [23] in their ODIN method, we improved ODIN's AUROC/AUPR-In/AUPR-Out results from 92.1/89.0/93.6 to 95.7/96.1/95.4, respectively. It is important to note that in contrast to our method, ODIN is *inapplicable* in the pure single class setting, where there are no class labels.

## 6 On the Intuition for Using Geometric Transformations

In this section we explain our intuition behind the choice of the set of transformations used in our method. Any bijection of a set (having some geometric structure) to itself is a *geometric transformation*. Among all geometric transformations, we only used compositions of *horizontal flipping, translations*, and *rotations* in our model, resulting in 72 distinct transformations (see supplementary material for the entire list). In the earlier stages of this work, we tried a few non-geometric transformations (e.g., Gaussian blur, sharpening, gamma correction), which degraded performance and we abandoned them altogether. We hypothesize that non-geometric transformations perform worse since they can eliminate important features of the learned image set.

We speculate that the effectiveness of the chosen transformation set is affected by their ability to preserve spatial information about the given "normal" images, as well as the ability of our classifier to predict which transformation was applied on a given transformed image. In addition, for a fixed type-II error rate, the type-I error rate of our method decreases the harder it gets for the trained classifier to correctly predict the identity of the transformations that were applied on anomalies.

We demonstrate this idea by conducting three experiments. Each experiment has the following structure. We train a neural classifier to discriminate between two transformations, where the normal class is taken to be images of a single digit from the MNIST [22] training set. We then evaluate our method using AUROC on a set of images comprising normal images and images of another digit from the MNIST test set. The three experiments are:

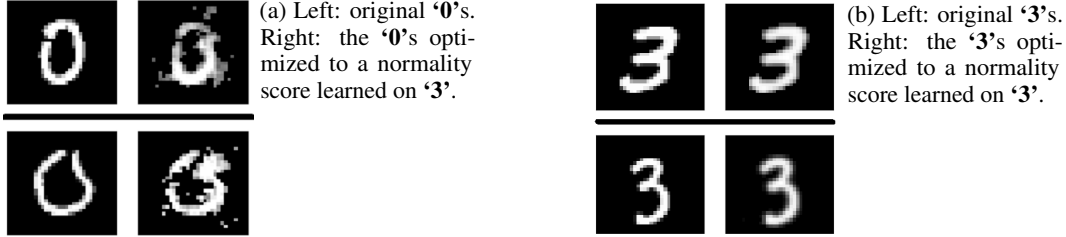

(a) Left: original **'0'**s. Right: the **'0'**s optimized to a normality score learned on **'3'**.

(b) Left: original **'3'**s. Right: the **'3'**s optimized to a normality score learned on **'3'**.

Figure 1: Optimizing digit images to maximize the normality score

- **Normal digit: '8'. Anomaly: '3'. Transformations: Identity and horizontal flip.** It can be expected that due to the invariance of '8' to horizontal flip, the classifier will have difficulties learning distinguishing features. Indeed, when presented with the test set containing '3' as anomalies (which do not exhibit such invariance), our method did not perform well, achieving an AUROC of 0.646.

- **Normal digit: '3'. Anomaly: '8'. Transformations: Identity and horizontal flip.** In contrast to the previous experiment, the transformed variants of digit '3' can easily be classified to the correct transformation. Indeed, our method, using the trained model for '3', achieved 0.957 AUROC in this experiment.

- **Normal digit: '8'. Anomaly: '3'. Transformations: Identity and translation by 7 pixels.** In this experiment, the transformed images are distinguishable from each other. As can be expected, our method performs well in this case, achieving an AUROC of 0.919.

To convince ourselves that high scores given by our scoring function indicate membership in the normal class, we tested how an image would need to change in order to obtain a high normality score. This was implemented by optimizing an input image using gradient ascent to maximize the simplified variant of the normality score described in section 5.5 (see, e.g., [39]). Thus, we trained a classifier on the digit '3' from the MNIST dataset, with a few geometric transformations. We then took an arbitrary image of the digit '0' and optimized it. In Figure 1(a) we present two such images, where the left one is the original, and the right is the result after taking 200 gradient ascent steps that "optimize" the original image. It is evident that the '0' digits have deformed, now resembling the digit '3'. This illustrates the fact that the classification model has learned features relevant to the "normal" class. To further strengthen our hypothesis, we conducted the same experiment using images from the normal class (i.e., images of the digit '3'). We expected these images to maintain their appearance during the optimization process, since they already contain the features that should contribute to a high normality score. Figure 1(b) contains two examples of the process, where in each row, the left image is the initial '3', and the right is the result after taking 200 gradient ascent steps on it. As hypothesized, it is evident that the images remained roughly unchanged at the end of the optimization process (regardless of their different orientations).

# 7 Conclusion and Future Work

We presented a novel method for anomaly detection of images, which learns a meaningful representation of the learned training data in a fully discriminative fashion. The proposed method is computationally efficient, and as simple to implement as a multi-class classification task. Unlike best-known methods so far, our approach completely alleviates the need for a generative component (autoencoders/GANs). Most importantly, our method significantly advances the state-of-the-art by offering a dramatic improvement over the best available anomaly detection methods. Our results open many avenues for future research. First, it is important to develop a theory that grounds the use of geometric transformations. It would be interesting to study the possibility of selecting transformations that would best serve a given training set, possibly with prior knowledge on the anomalous samples. Another avenue is explicitly optimizing the set of transformations. Due to the effectiveness of our method, it is tempting to try adapting it to other settings or utilizing it in applications. Some examples are open-world classification, selective classification and regression [36, 35, 13], uncertainty estimation [15], and deep active learning [12, 14]. Finally, it would be interesting to consider using our techniques in settings where additional unlabeled "contaminated" data (consisting of both normal and novel instances) is provided, perhaps within a transductive learning framework [10].

Table 1: Average area under the ROC curve in % with SEM (over 5 runs) of anomaly detection methods. For all datasets, each model was trained on the single class, and tested against all other classes. E2E column is taken from [27]. OC-SVM hyperparameters in RAW and CAE variants were optimized with hindsight knowledge. The best performing method in each experiment is in bold.

| Dataset | $c_i$ | OC-SVM | | | DAGMM | DSEBM | AD-GAN | OURS |
|---|---|---|---|---|---|---|---|---|
| | | RAW | CAE | E2E | | | | |
| CIFAR-10 (32x32x3) | 0 | 70.6 | **74.9** | 61.7±1.3 | 41.4±2.3 | 56.0±6.9 | 64.9 | 74.7±0.4 |
| | 1 | 51.3 | 51.7 | 65.9±0.7 | 57.1±2.0 | 48.3±1.8 | 39.0 | **95.7±0.0** |
| | 2 | 69.1 | 68.9 | 50.8±0.3 | 53.8±4.0 | 61.9±0.1 | 65.2 | **78.1±0.4** |
| | 3 | 52.4 | 52.8 | 59.1±0.4 | 51.2±0.8 | 50.1±0.4 | 48.1 | **72.4±0.5** |
| | 4 | 77.3 | 76.7 | 60.9±0.3 | 52.2±7.3 | 73.3±0.2 | 73.5 | **87.8±0.2** |
| | 5 | 51.2 | 52.9 | 65.7±0.8 | 49.3±3.6 | 60.5±0.3 | 47.6 | **87.8±0.1** |
| | 6 | 74.1 | 70.9 | 67.7±0.8 | 64.9±1.7 | 68.4±0.3 | 62.3 | **83.4±0.5** |
| | 7 | 52.6 | 53.1 | 67.3±0.3 | 55.3±0.8 | 53.3±0.7 | 48.7 | **95.5±0.1** |
| | 8 | 70.9 | 71.0 | 75.9±0.4 | 51.9±2.4 | 73.9±0.3 | 66.0 | **93.3±0.0** |
| | 9 | 50.6 | 50.6 | 73.1±0.4 | 54.2±5.8 | 63.6±3.1 | 37.8 | **91.3±0.1** |
| | *avg* | 62.0 | 62.4 | 64.8 | 53.1 | 60.9 | 55.3 | **86.0** |
| CIFAR-100 (32x32x3) | 0 | 68.0 | 68.4 | - | 43.4±3.9 | 64.0±0.2 | 63.1 | **74.7±0.4** |
| | 1 | 63.1 | 63.6 | - | 49.5±2.7 | 47.9±0.1 | 54.9 | **68.5±0.2** |
| | 2 | 50.4 | 52.0 | - | 66.1±1.7 | 53.7±4.1 | 41.3 | **74.0±0.5** |
| | 3 | 62.7 | 64.7 | - | 52.6±1.0 | 48.4±0.5 | 50.0 | **81.0±0.8** |
| | 4 | 59.7 | 58.2 | - | 56.9±3.0 | 59.7±6.3 | 40.6 | **78.4±0.5** |
| | 5 | 53.5 | 54.9 | - | 52.4±2.2 | 46.6±1.6 | 42.8 | **59.1±1.0** |
| | 6 | 55.9 | 57.2 | - | 55.0±1.1 | 51.7±0.8 | 51.1 | **81.8±0.2** |
| | 7 | 64.4 | 62.9 | - | 52.8±3.7 | 54.8±1.6 | 55.4 | **65.0±0.1** |
| | 8 | 66.7 | 65.6 | - | 53.2±4.8 | 66.7±0.2 | 59.2 | **85.5±0.4** |
| | 9 | 70.1 | 74.1 | - | 42.5±2.5 | 71.2±1.2 | 62.7 | **90.6±0.1** |
| | 10 | 83.0 | 84.1 | - | 52.7±3.9 | 78.3±1.1 | 79.8 | **87.6±0.2** |
| | 11 | 59.7 | 58.0 | - | 46.4±2.4 | 62.7±0.7 | 53.7 | **83.9±0.6** |
| | 12 | 68.7 | 68.5 | - | 42.7±3.1 | 66.8±0.0 | 58.9 | **83.2±0.3** |
| | 13 | **65.0** | 64.6 | - | 45.4±0.7 | 52.6±0.1 | 57.4 | 58.0±0.4 |
| | 14 | 50.7 | 51.2 | - | 57.2±1.3 | 44.0±0.6 | 39.4 | **92.1±0.2** |
| | 15 | 63.5 | 62.8 | - | 48.8±1.5 | 56.8±0.1 | 55.6 | **68.3±0.1** |
| | 16 | 68.3 | 66.6 | - | 54.4±3.1 | 63.1±0.1 | 63.3 | **73.5±0.2** |
| | 17 | 71.7 | 73.7 | - | 36.4±2.3 | 73.0±1.0 | 66.7 | **93.8±0.1** |
| | 18 | 50.2 | 52.8 | - | 52.4±1.4 | 57.7±1.6 | 44.3 | **90.7±0.1** |
| | 19 | 57.5 | 58.4 | - | 50.3±1.0 | 55.5±0.7 | 53.0 | **85.0±0.2** |
| | *avg* | 62.6 | 63.1 | - | 50.5 | 58.8 | 54.7 | **78.7** |
| Fashion-MNIST (32x32x1) | 0 | 98.2 | 97.7 | - | 42.1±9.1 | 91.6±1.2 | 89.9 | **99.4±0.0** |
| | 1 | 90.3 | 89.9 | - | 55.1±3.5 | 71.8±0.5 | 81.9 | **97.6±0.1** |
| | 2 | 90.7 | **91.4** | - | 50.4±7.3 | 88.3±0.2 | 87.6 | 91.1±0.2 |
| | 3 | **94.2** | 90.7 | - | 57.0±6.7 | 87.3±3.6 | 91.2 | 89.9±0.4 |
| | 4 | 89.4 | 89.1 | - | 26.9±5.4 | 85.2±0.9 | 86.5 | **92.1±0.0** |
| | 5 | 91.8 | 88.5 | - | 70.5±9.7 | 87.1±0.0 | 89.6 | **93.4±0.9** |
| | 6 | **83.4** | 81.7 | - | 48.3±5.0 | 73.4±4.1 | 74.3 | 83.3±0.1 |
| | 7 | 98.8 | 98.7 | - | 83.5±11.4 | 98.1±0.0 | 97.2 | **98.9±0.1** |
| | 8 | **91.9** | 90.6 | - | 49.9±7.2 | 86.0±3.2 | 89.0 | 90.8±0.1 |
| | 9 | 99.0 | 98.6 | - | 34.0±3.0 | 97.1±0.3 | 97.1 | **99.2±0.0** |
| | *avg* | 92.8 | 91.7 | - | 51.8 | 86.6 | 88.4 | **93.5** |
| CatsVsDogs (64x64x3) | 0 | 50.4 | 55.2 | - | 43.4±0.5 | 47.1±1.7 | 50.7 | **88.3±0.3** |
| | 1 | 53.0 | 49.9 | - | 52.0±1.9 | 56.1±1.2 | 48.1 | **89.2±0.3** |
| | *avg* | 51.7 | 52.5 | - | 47.7 | 51.6 | 49.4 | **88.8** |

## Acknowledgements

This research was partially supported by the Israel Science Foundation (grant No. 710/18).

## Footnotes

[1] Unless otherwise mentioned, the use of the adjective "normal" is unrelated to the Gaussian distribution.

[2]A complete code of the proposed method's implementation and the conducted experiments is available at `https://github.com/izikgo/AnomalyDetectionTransformations`.

[3]The parameters 16, 8 were used on CIFAR-10 by the authors. Due to the induced computational complexity, we chose smaller values. When testing the parameters 16, 8 with our method on the CIFAR-10 dataset, anomaly detection results improved.

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
