[Supplementary Material · NIPS_2018_supp_mat.pdf]

# Deep Anomaly Detection Using Geometric Transformations – Supplementary Material

**Izhak Golan**
Department of Computer Science
Technion – Israel Institute of Technology
Haifa, Israel
izikgo@cs.technion.ac.il

**Ran El-Yaniv**
Department of Computer Science
Technion – Israel Institute of Technology
Haifa, Israel
rani@cs.technion.ac.il

## A    List of Geometric Transformations Used By Our Method

In all our experiments, except for those described in Section 6 we used a fixed set of 72 geometric transformations. These transformations can be succinctly described as the composition of the following transformations, applied on an image in the order they are listed:

- **Horizontal flip**: denoted as $T_b^{flip}(x)$, where $b \in \{T, F\}$. The parameter $b$ indicates whether the flipping occurs, or the transformation is the identity.

- **Translation**: denoted as $T_{s_h,s_w}^{trans}(x)$, where $s_h, s_w \in \{-1, 0, 1\}$. Applying this transformation on an image translates it by 0.25 of its height, and 0.25 of its width, in both dimensions. The direction of the translation in each axis is determined by $s_h$ and $s_w$, where $s_h = s_w = 0$ means no translation. A reflection is used to complete missing pixels.

- **Rotation by multiples 90 degrees**: denoted as $T_k^{rot}(x)$, where $k \in \{0, 1, 2, 3\}$. Applying this transformation on an image rotates it counter-clockwise by $k \times 90$ degrees.

The entire set of transformations is thus given by

$$\mathcal{T} = \left\{ T_k^{rot} \circ T_{s_h,s_w}^{trans} \circ T_b^{flip} : \begin{matrix} b \in \{T, F\}, \\ s_h, s_w \in \{-1, 0, 1\}, \\ k \in \{0, 1, 2, 3\} \end{matrix} \right\}$$

By taking all possible compositions, we obtain a total of $2 \times 3 \times 3 \times 4 = 72$ transformations, where each composition is fully defined by a tuple, $(b, s_w, s_h, k)$. For example, the identity transformation is $(F, 0, 0, 0)$.

## B    Algorithm

We present here a full and detailed algorithm of our deep anomaly detection technique. The function $\Psi(\cdot)$ is the Digamma function, and its inverse is calculated numerically using five Newton-Raphson iterations.

---

**Algorithm 1** Deep Anomaly Detection Using Geometric Transformations

---

     **Input:** $S$: a set of "normal" images. $\mathcal{T} = \{T_0, T_1, \ldots, T_{k-1}\}$: a set of geometric transformations.
     $f_\theta$: a softmax classifier parametrized by $\theta$.
     **Output:** A normality scoring function $n_S(x)$.

1:  **procedure** GETNORMALITYSCORE$(S, \mathcal{T}, f_\theta)$
2:     $S_\mathcal{T} \leftarrow \{(T_j(x), j) : x \in S, T_j \in \mathcal{T}\}$
3:     **while** not converged **do**
4:        Train $f_\theta$ on the labeled set $S_\mathcal{T}$
5:     **end while**
6:     $n \leftarrow |S|$
7:     **for** $i \in \{0, \ldots, k-1\}$ **do**
8:        $S_i \leftarrow \{\mathbf{y}(T_i(x)) | x \in S\}$
9:        $\bar{\boldsymbol{s}} \leftarrow \frac{1}{n} \sum_{\boldsymbol{s} \in S_i} \boldsymbol{s}$
10:       $\bar{\boldsymbol{l}} \leftarrow \frac{1}{n} \sum_{\boldsymbol{s} \in S_i} \log \boldsymbol{s}$
11:       $\tilde{\boldsymbol{\alpha}}_i \leftarrow \bar{\boldsymbol{s}} \frac{(k-1)(-\Psi(1))}{\bar{\boldsymbol{s}} \cdot \log \bar{\boldsymbol{s}} - \bar{\boldsymbol{s}} \cdot \bar{\boldsymbol{l}}}$                        ▷ Initialization from [31]
12:       **while** not converged **do**
13:          $\tilde{\boldsymbol{\alpha}}_i \leftarrow \Psi^{-1}\left(\Psi\left(\sum_j [\tilde{\boldsymbol{\alpha}}_i]_j\right) + \bar{\boldsymbol{l}}\right)$         ▷ Fixed point method from [21]
14:       **end while**
15:     **end for**
16:     **return** $n_S(x) \triangleq \sum_{i=0}^{k-1} (\tilde{\boldsymbol{\alpha}}_i - 1) \cdot \log \mathbf{y}(T_i(x))$
17: **end procedure**

---

# C  Single Class Names

The following table describes the content of all single classes.

| Dataset | $c_i$ | Single Class Name |
| --- | --- | --- |
| CIFAR-10 | 0 | Airplane |
| | 1 | Automobile |
| | 2 | Bird |
| | 3 | Cat |
| | 4 | Deer |
| | 5 | Dog |
| | 6 | Frog |
| | 7 | Horse |
| | 8 | Ship |
| | 9 | Truck |
| CIFAR-100 | 0 | Aquatic mammals |
| | 1 | Fish |
| | 2 | Flowers |
| | 3 | Food containers |
| | 4 | Fruit and vegetables |
| | 5 | Household electrical devices |
| | 6 | Household furniture |
| | 7 | Insects |
| | 8 | Large carnivores |
| | 9 | Large man-made outdoor things |
| | 10 | Large natural outdoor scenes |
| | 11 | Large omnivores and herbivores |
| | 12 | Medium-sized mammals |
| | 13 | Non-insect invertebrates |
| | 14 | People |
| | 15 | Reptiles |
| | 16 | Small mammals |
| | 17 | Trees |
| | 18 | Vehicles 1 |
| | 19 | Vehicles 2 |
| Fashion-MNIST | 0 | Ankle-boot |
| | 1 | Bag |
| | 2 | Coat |
| | 3 | Dress |
| | 4 | Pullover |
| | 5 | Sandal |
| | 6 | Shirt |
| | 7 | Sneaker |
| | 8 | T-shirt |
| | 9 | Trouser |
| CatsVsDogs | 0 | Cat |
| | 1 | Dog |

## D    Examples from the Fashion-MNIST Dataset

## E    Area Under the Precision-Recall Curve

When dealing with highly skewed datasets, precision-recall curves may be considered more informative than the area under the ROC. For completeness, we provide results of all baseline models in terms of area under the precision-recall curve. This score can be calculated in two ways: anomalies treated as the positive class (AUPR-out), and anomalies treated as the negative class (AUPR-in).

Table 1: Average area under the PR curve in % with SEM (computed over 5 runs) of anomaly detection methods, when anomalies are taken as the negative class (AUPR-In). For all datasets, each model was trained on the single class, and tested against all other classes. The best performing method in each experiment is in bold.

| Dataset | $c_i$ | OC-SVM | | DAGMM | DSEBM | AD-GAN | OURS |
|---|---|---|---|---|---|---|---|
| | | RAW | CAE | | | | |
| CIFAR-10 (32x32x3) | 0 | 25.6 | **33.2** | 8.3±0.5 | 14.4±2.8 | 17.9 | 28.9±0.6 |
| | 1 | 56.2 | 56.5 | 12.5±0.8 | 9.3±0.5 | 8.2 | **78.7±0.3** |
| | 2 | 23.0 | 24.0 | 12.5±1.1 | 18.5±0.1 | 18.0 | **33.1±1.4** |
| | 3 | 11.2 | 12.1 | 10.2±0.2 | 9.7±0.1 | 9.0 | **33.2±0.8** |
| | 4 | 27.9 | 29.2 | 12.5±2.2 | 23.0±0.2 | 21.5 | **42.9±0.3** |
| | 5 | 9.7 | 12.2 | 9.9±0.9 | 12.7±0.4 | 9.1 | **52.9±0.5** |
| | 6 | 20.0 | 21.7 | 17.2±1.3 | 17.6±0.0 | 13.6 | **48.8±2.1** |
| | 7 | 11.6 | 12.4 | 11.4±0.3 | 10.7±0.7 | 9.0 | **81.3±0.2** |
| | 8 | 25.6 | 25.7 | 13.7±1.7 | 27.1±1.3 | 16.5 | **68.4±0.3** |
| | 9 | 40.9 | 48.5 | 12.4±2.3 | 17.4±2.5 | 7.3 | **58.6±0.4** |
| | $avg$ | 25.2 | 27.6 | 12.1 | 16.1 | 13.0 | **52.7** |
| CIFAR-100 (32x32x3) | 0 | 11.7 | **14.1** | 4.5±0.5 | 8.9±0.1 | 7.1 | 14.0±0.3 |
| | 1 | 12.9 | **14.9** | 5.8±0.9 | 5.1±0.0 | 5.8 | 12.5±0.3 |
| | 2 | 4.7 | 4.9 | 9.6±1.0 | 5.9±1.1 | 4.2 | **16.0±0.7** |
| | 3 | 15.4 | 21.5 | 5.5±0.3 | 5.0±0.1 | 5.1 | **41.2±1.0** |
| | 4 | 10.8 | 17.9 | 6.5±1.1 | 9.7±2.4 | 3.8 | **30.5±1.2** |
| | 5 | 7.1 | **11.2** | 5.7±0.5 | 4.6±0.2 | 4.1 | 8.8±0.3 |
| | 6 | 6.2 | 7.5 | 6.3±0.5 | 5.2±0.1 | 5.1 | **36.1±1.0** |
| | 7 | 9.0 | **13.0** | 5.8±0.7 | 7.6±0.7 | 6.7 | 9.3±0.1 |
| | 8 | 7.8 | 7.8 | 5.9±0.9 | 7.9±0.1 | 5.7 | **36.2±0.9** |
| | 9 | 9.4 | 11.9 | 4.0±0.2 | 10.6±0.9 | 6.7 | **39.8±0.5** |
| | 10 | 28.1 | 32.1 | 6.3±0.7 | 27.8±1.2 | 19.5 | **41.0±0.8** |
| | 11 | 6.3 | 6.0 | 4.6±0.3 | 7.0±0.2 | 5.3 | **32.1±1.4** |
| | 12 | 9.4 | 10.3 | 4.1±0.3 | 8.4±0.0 | 6.6 | **32.0±0.7** |
| | 13 | 12.6 | **15.2** | 4.6±0.2 | 6.3±0.1 | 9.1 | 6.7±0.1 |
| | 14 | 15.2 | 53.6 | 6.1±0.4 | 4.1±0.1 | 3.7 | **63.6±0.5** |
| | 15 | 8.9 | 10.3 | 5.0±0.3 | 7.5±0.1 | 6.4 | **10.4±0.1** |
| | 16 | 12.8 | **15.6** | 6.1±0.5 | 8.1±0.0 | 8.3 | 13.4±0.2 |
| | 17 | 10.0 | 12.1 | 3.7±0.1 | 14.3±2.9 | 7.3 | **66.4±0.7** |
| | 18 | 4.7 | 5.1 | 5.4±0.3 | 6.4±0.4 | 3.8 | **44.8±0.3** |
| | 19 | 7.3 | 10.6 | 5.1±0.3 | 5.7±0.2 | 5.2 | **26.7±0.3** |
| | $avg$ | 10.5 | 14.8 | 5.5 | 8.3 | 6.5 | **29.1** |
| Fashion-MNIST (32x32x1) | 0 | 86.1 | 84.7 | 9.5±3.5 | 55.3±7.6 | 41.7 | **95.1±0.4** |
| | 1 | 62.2 | 75.3 | 12.5±2.5 | 22.2±1.7 | 27.2 | **91.8±0.5** |
| | 2 | 53.4 | **58.7** | 11.7±4.0 | 41.3±0.5 | 39.7 | 46.2±0.8 |
| | 3 | **70.4** | 67.8 | 13.9±4.9 | 47.6±7.8 | 60.1 | 53.8±1.8 |
| | 4 | 53.4 | 51.4 | 6.0±0.2 | 35.8±0.8 | 38.8 | **54.1±0.4** |
| | 5 | 58.7 | 60.1 | 39.2±11.0 | 35.8±0.1 | 51.8 | **63.0±4.5** |
| | 6 | 35.3 | **42.4** | 18.9±9.1 | 23.3±2.4 | 25.7 | 30.3±0.7 |
| | 7 | 91.3 | **92.2** | 44.5±10.2 | 86.9±0.5 | 80.3 | 87.5±1.8 |
| | 8 | 67.0 | **69.4** | 9.6±1.7 | 49.8±7.6 | 55.4 | 53.9±0.8 |
| | 9 | 96.1 | 95.6 | 5.4±0.1 | 88.8±1.3 | 90.5 | **97.1±0.1** |
| | $avg$ | 67.4 | **69.8** | 17.1 | 48.7 | 51.1 | 67.3 |
| CatsVsDogs (64x64x3) | 0 | 50.5 | 54.6 | 45.4±0.3 | 49.0±1.3 | 51.0 | **90.6±0.3** |
| | 1 | 52.1 | 40.7 | 56.7±4.7 | 54.0±1.0 | 48.6 | **91.1±0.2** |
| | $avg$ | 51.3 | 47.6 | 51.1 | 51.5 | 49.8 | **90.9** |

Table 2: Average area under the PR curve in % with SEM (computed over 5 runs) of anomaly detection methods, when anomalies are taken as the positive class (AUPR-Out). For all datasets, each model was trained on the single class, and tested against all other classes. The best performing method in each experiment is in bold.

| Dataset | $c_i$ | OC-SVM RAW | OC-SVM CAE | DAGMM | DSEBM | AD-GAN | OURS |
|---|---|---|---|---|---|---|---|
| CIFAR-10 (32x32x3) | 0 | 95.2 | **95.7** | 88.0±0.9 | 91.3±1.6 | 93.4 | 95.4±0.1 |
| | 1 | 95.1 | 95.2 | 91.9±0.6 | 89.7±0.4 | 86.3 | **99.5±0.0** |
| | 2 | 94.8 | 95.2 | 90.4±1.3 | 92.1±0.0 | 94.0 | **96.4±0.1** |
| | 3 | 91.6 | 91.0 | 90.6±0.3 | 90.3±0.1 | 90.2 | **95.0±0.1** |
| | 4 | 96.4 | 96.4 | 90.2±2.1 | 95.6±0.0 | 96.0 | **98.2±0.0** |
| | 5 | 90.8 | 91.2 | 90.1±1.0 | 93.2±0.1 | 90.0 | **98.2±0.0** |
| | 6 | 96.0 | 95.3 | 93.6±0.4 | 94.2±0.1 | 93.2 | **97.2±0.1** |
| | 7 | 91.9 | 92.0 | 91.9±0.2 | 91.4±0.1 | 90.4 | **99.4±0.0** |
| | 8 | 95.0 | 95.1 | 89.4±0.7 | 95.2±0.1 | 94.3 | **99.1±0.0** |
| | 9 | 95.0 | 95.1 | 91.8±1.4 | 93.4±0.6 | 87.3 | **98.8±0.0** |
| | *avg* | 94.2 | 94.2 | 90.8 | 92.6 | 91.5 | **97.7** |
| CIFAR-100 (32x32x3) | 0 | 97.3 | 97.3 | 93.9±0.6 | 96.9±0.0 | 97.0 | **98.0±0.0** |
| | 1 | 96.9 | 96.8 | 94.7±0.5 | 94.8±0.0 | 95.9 | **97.4±0.0** |
| | 2 | 95.4 | 95.7 | 97.3±0.1 | 95.8±0.5 | 93.8 | **97.8±0.0** |
| | 3 | 96.8 | 96.7 | 95.2±0.1 | 94.4±0.1 | 94.7 | **98.2±0.1** |
| | 4 | 96.3 | 96.0 | 96.4±0.3 | 96.4±0.7 | 93.9 | **98.2±0.1** |
| | 5 | **97.3** | 95.3 | 95.1±0.3 | 94.2±0.2 | 93.9 | 95.9±0.1 |
| | 6 | 95.9 | 96.3 | 96.1±0.2 | 95.4±0.2 | 95.0 | **98.5±0.0** |
| | 7 | 96.9 | 96.7 | 95.4±0.5 | 95.1±0.3 | 95.6 | **96.9±0.0** |
| | 8 | 97.4 | 97.3 | 95.6±0.6 | 97.4±0.0 | 96.8 | **99.0±0.0** |
| | 9 | 97.8 | 98.1 | 94.5±0.4 | 97.8±0.1 | 97.1 | **99.4±0.0** |
| | 10 | 98.8 | 98.9 | 95.0±0.4 | 98.3±0.1 | 98.6 | **99.1±0.0** |
| | 11 | 96.7 | 96.5 | 94.6±0.3 | 97.0±0.1 | 95.9 | **98.8±0.0** |
| | 12 | 97.4 | 97.4 | 94.2±0.5 | 97.4±0.0 | 96.3 | **98.7±0.0** |
| | 13 | **97.0** | 96.7 | 94.2±0.2 | 94.9±0.0 | 95.4 | 96.1±0.0 |
| | 14 | 97.5 | 97.6 | 96.2±0.1 | 94.7±0.1 | 94.0 | **99.4±0.0** |
| | 15 | 97.0 | 96.9 | 94.5±0.2 | 95.4±0.0 | 95.6 | **97.4±0.0** |
| | 16 | 97.4 | 97.1 | 95.5±0.4 | 96.8±0.0 | 96.8 | **97.9±0.0** |
| | 17 | 97.9 | 98.1 | 93.9±0.5 | 97.9±0.1 | 97.6 | **99.6±0.0** |
| | 18 | 95.5 | 95.8 | 95.7±0.2 | 96.3±0.1 | 94.8 | **99.4±0.0** |
| | 19 | 96.5 | 96.5 | 95.2±0.2 | 95.7±0.1 | 95.7 | **98.9±0.0** |
| | *avg* | 97.0 | 96.9 | 95.2 | 96.1 | 95.7 | **98.2** |
| Fashion-MNIST (32x32x1) | 0 | 99.8 | 99.7 | 91.3±2.1 | 98.9±0.1 | 98.9 | **99.8±0.0** |
| | 1 | 98.6 | 98.2 | 94.0±0.4 | 95.6±0.1 | 97.4 | **99.6±0.0** |
| | 2 | 98.8 | 98.8 | 92.9±1.4 | 98.5±0.1 | 98.4 | **99.0±0.0** |
| | 3 | **99.3** | 98.7 | 95.0±0.7 | 98.4±0.5 | 98.8 | 98.7±0.0 |
| | 4 | 98.6 | 98.6 | 85.4±2.4 | 98.0±0.1 | 98.2 | **99.1±0.0** |
| | 5 | 99.0 | 98.4 | 96.6±1.1 | 98.4±0.0 | 98.8 | **99.2±0.1** |
| | 6 | 97.8 | 97.3 | 91.2±1.7 | 95.9±0.9 | 96.0 | **97.9±0.0** |
| | 7 | 99.9 | 99.8 | 98.0±1.4 | 99.8±0.0 | 99.7 | **99.9±0.0** |
| | 8 | 98.8 | 98.5 | 92.2±1.4 | 98.0±0.5 | 98.5 | **98.9±0.0** |
| | 9 | 99.9 | 99.8 | 91.1±0.8 | 99.6±0.0 | 99.6 | **99.9±0.0** |
| | *avg* | 99.0 | 98.8 | 92.8 | 98.1 | 98.4 | **99.2** |
| CatsVsDogs (64x64x3) | 0 | 57.2 | 53.5 | 46.3±0.4 | 47.3±1.1 | 50.0 | **83.3±0.4** |
| | 1 | 53.3 | 74.9 | 58.2±4.8 | 55.8±1.1 | 48.6 | **85.4±0.5** |
| | *avg* | 55.3 | 64.2 | 52.2 | 51.5 | 49.3 | **84.3** |