[Reviews · NeurIPS 2018]

Reviewer 1



The authors proposed an original approach to image anomaly detection. They proposed to define a set of geometric transformations and then to construct a classifier to predict to which type of transformation a given image is more similar. In case the classifier score is low, the image is considered as anomalous. As a test set the authors considered standard image datasets like CIFAR, Fashion-MNIST and CatsVsDogs. However, there exist special datasets to evaluate anomaly detection methods for images. E.g. http://odds.cs.stonybrook.edu/#table5 The UCSD anomaly detection is annotated: each video frame is annotated as anomalous/non-anomalous and sometimes it contains pixel-wise annotations. Such datasets are closer to real-life applications and should be used for testing. As baselines the authors used - one-class SVM based either on pixels, or on low-dimension representation from the neural network - energy-based models - deep autoencoding GMM - GANs The authors claimed that in case of one-class SVM they used the best hyperparameters as there is no approach how to tune them. In fact, this is not true. There are approaches how to tune hyperparameters of one-class SVM: - Wang, S., Liu, Q., et al.: Hyperparameter selection of one-class support vector machine by self-adaptive data shifting. Pattern Recognition 74 (2018) 198–211 - Thomas, A., Feuillard, V., et al.: Calibration of one-class svm for mv set estimation. 
CoRR abs/1508.07535 (2015) - E. Burnaev, P. Erofeev, D. Smolyakov. Model Selection for Anomaly Detection // Proc. SPIE 9875, Eighth International Conference on Machine Vision, 987525 (December 8, 2015); 5 P. doi:10.1117/12.2228794; http://dx.doi.org/10.1117/12.2228794 Moreover, additional gain in accuracy when using one-class SVM can be obtained through end-to-end learning of embeddings specifically for one-class classification. There exists an approach how to do that, namely, https://arxiv.org/pdf/1804.04888.pdf Also see https://arxiv.org/pdf/1801.05365.pdf Of course, the one-class SVM is not the most powerful approach for anomaly detection. Experiments on numerous benchmark datasets with ground truth that compared popular anomaly detection algorithms find that One Class SVM ranks at the bottom, see We note that the top performer in [1] is the Isolation Forest algorithm [2], an ensemble of randomized trees. Thus it is important to include in the comparison Isolation Forest algorithm based on the same features, as one-class SVM! [1] Emmott, A., Das, S., et al.: Systematic construction of anomaly detection bench-marks from real data. In: Proceedings of KDD ODD. (2013) [2] Liu, F., Ting, K., Zhou, Z.H.: Isolation forest. In: Proceedings of ICDM. (2008) - In general, features, generated by neural networks, are believed to provide efficient characterization of image data. In particular, very often it is enough to use linear classifier in the feature space, produce by deep layers of the neural network. Thus, why did the authors use kernel one-class SVM (CAE-OC-SVM)? It seems that it can be enough to use linear one-class SVM with features, produced by usual deep network such as VGG. - another issue concerning section 5.1. PCA should be used as a baseline! It is very often used in image processing engineering applications, including anomaly detection. - page 3, line 104. The authors proposed to use AUC. However, this may be not appropriate measure to estimate performance of anomaly detector. In fact, in real datasets the number of anomalies is small compareв to the size of the normal class. Thus area under the precision-recall curve is better suited for such imbalanced cases. - displayed formula after the line 100: after the first line of the displayed formula a comma sign should be used - The idea to use some class of geometric transformations T looks similar to the idea of image set augmentation. It is obvious that “richness” of the used class of geometric transformations should significantly influence anomaly detection accuracy. However, the authors did not investigate how anomaly detection accuracy depend on the diversity of the used class of geometric transformations. To which extent obtained anomaly detection results are sensitive to properties of this class? - Practical image classification problems often contain a lot of classes (so-called extreme classification problems). Is the proposed method capable to deal with such case? To which extent obtained anomaly detection performance is sensitive to the number of classes, used to model anomalous observations? - in section 5.3 the authors consider issues about hyperparameters. Are there any recommendations how to select a particular architecture of the neural network? Should we try to find such architecture which provides as accurate classification of types of image transformations as possible? - in the paper by Liang et al. “Enhancing the reliability …” they proposed the method that using temperature scaling and adding small perturbations to the input can separate the softmax score distributions between in- and out-of-distribution images, allowing for effective detection of out-of-distribution images. In fact, in the current paper the authors proposed to use a set of geometric transformation as such perturbations. Conclusion: - the topic of the paper is important - the paper is well-written. However, still there are a number of open questions, see comments above. - the results are novel. However, the novelty is not very big compared, e.g. to the results of the paper by Liang et al.

Reviewer 2



The authors present a framework for image anomaly detection. This method seems to advance the state-of-art by learning a scoring function ns_(x) from which a classifier can be constructed with an anomaly threshold. They also did experiments to demonstrate the idea about causes of the effectiveness. The authors has verified the performance of their method on sufficient and diverse databases , but the approaches which they compare with are not very new to me, it may be more convincing if the experiments can be conducted by comparing with the latest methods, e.g., #1 Hendrycks, Dan, and Kevin Gimpel. "A baseline for detecting misclassified and out-of-distribution examples in neural networks." International Conference on Learning Representations (ICLR), 2017. #2 Liang, Shiyu, Yixuan Li, and R. Srikant. "Enhancing the reliability of out-of-distribution image detection in neural networks." International Conference on Learning Representations (ICLR), 2018. The authors also analyze the reason why their method are not outstanding on the same datasets, but the method DAGMM they compare with is unsupervised while their method in this paper seems supervised to me. In this sense, this comparison may not be very appropriate to me, or it should be further clarified at least. The rebuttal seems to have addressed my major concerns in the experiments.

Reviewer 3



This work focuses on detecting anomalies in images by identifying out-of-distribution images. Their approach is to train a multiclass model to classify different geometric transformations of training images. They claim that such a model can generate feature representation that is useful for detecting novelties. Given a set of images, they apply a set of geometric transformations on each of these images. Next, a model is created to classify the transformed images based on classification. During testing, each of the transformation is applied to the test image and passed through the trained model to produce a softmax output. The score of the input image is the mean softmax output of all the transformations. They compare the performance of their approach against a variety of baselines on different datsets and show that their method can work better. It seemed a bit confusing when the work talks about classes in the main paper. There are 2 types of classes mentioned. 1) The set of geometric transformations, and 2) The classes inherently made available through supervised labeling of the data set itself. It seems that if there are k1 types of transformations and k2 classes in the labeled dataset (eg: k2 = 10 for CIFAR10) then anomaly detection is performed separately for each of the k2 classes. This might give substantial overhead as it seems that this requires training of k2 different models. What are the author's thoughts about directly training a model for k1 transformations using all the k2 classes ? What was the reason for using the 20 class grouping and not the 100 class version of the CIFAR 100 data? Do the number of classes (k2) in the dataset have an influence on performance? What does that mean for large datasets with a large number of classes. What about datasets that do not have class labels (as might be the case for real-world data). It will be interesting to know the influence of each geometric transformation on the performance. Furthermore, it seems that a very small set of transformations is utilized (some of which are common transformations used in data augmentation). Is there a specific reason for choosing these? Can fewer/more diverse transformations be used ?